# The Prognostic Value of *ASPHD1* and *ZBTB12* in Colorectal Cancer: A Machine Learning-Based Integrated Bioinformatics Approach

**DOI:** 10.3390/cancers15174300

**Published:** 2023-08-28

**Authors:** Alireza Asadnia, Elham Nazari, Ladan Goshayeshi, Nima Zafari, Mehrdad Moetamani-Ahmadi, Lena Goshayeshi, Haneih Azari, Ghazaleh Pourali, Ghazaleh Khalili-Tanha, Mohammad Reza Abbaszadegan, Fatemeh Khojasteh-Leylakoohi, MohammadJavad Bazyari, Mir Salar Kahaei, Elnaz Ghorbani, Majid Khazaei, Seyed Mahdi Hassanian, Ibrahim Saeed Gataa, Mohammad Ali Kiani, Godefridus J. Peters, Gordon A. Ferns, Jyotsna Batra, Alfred King-yin Lam, Elisa Giovannetti, Amir Avan

**Affiliations:** 1Metabolic Syndrome Research Center, Mashhad University of Medical Sciences, Mashhad 91779-48564, Iran; alirezaasadnia@gmail.com (A.A.); nima.zafri@gmail.com (N.Z.); mehrdadahmadi45@yahoo.com (M.M.-A.); azari.hanie@gmail.com (H.A.); ghazalehpourali@gmail.com (G.P.); ghazaleh.khalili24@gmail.com (G.K.-T.); fatemekhjst@gmail.com (F.K.-L.); ghorbanie971@mums.ac.ir (E.G.); khazaeim@mums.ac.ir (M.K.); hasanianmehrm@mums.ac.ir (S.M.H.); 2Medical Genetics Research Center, Mashhad University of Medical Sciences, Mashhad 91886-17871, Iran; abbaszadeganmr@mums.ac.ir (M.R.A.); salarkahaei1372@gmail.com (M.S.K.); 3Basic Sciences Research Institute, Mashhad University of Medical Sciences, Mashhad 13944-91388, Iran; kianima@mums.ac.ir; 4Department of Health Information Technology and Management, School of Allied Medical Sciences, Shahid Beheshti University of Medical Sciences, Tehran 19839-69411, Iran; nazarie4001@mums.ac.ir; 5Department of Gastroenterology and Hepatology, Faculty of Medicine, Mashhad University of Medical Sciences, Mashhad 91779-48564, Iran; ladangoshayeshi@gmail.com; 6Surgical Oncology Research Center, Mashhad University of Medical Sciences, Mashhad 91779-48954, Iran; goshayeshilena@gmail.com; 7Department of Medical Biotechnology, Faculty of Medicine, Mashhad University of Medical Sciences, Mashhad 91779-48564, Iran; bazyarimj981@mums.ac.ir; 8College of Medicine, University of Warith Al-Anbiyaa, Karbala 56001, Iraq; ibraheem@uowa.edu.iq; 9Department of Biochemistry, Medical University of Gdansk, 80-211 Gdansk, Poland; gj.peters@amsterdamumc.nl; 10Cancer Center Amsterdam, Amsterdam U.M.C., VU University Medical Center (VUMC), Department of Medical Oncology, 1081 HV Amsterdam, The Netherlands; 11Brighton & Sussex Medical School, Department of Medical Education, Falmer, Brighton, Sussex BN1 9PH, UK; g.ferns@bsms.ac.uk; 12Faculty of Health, School of Biomedical Sciences, Queensland University of Technology (QUT), Brisbane, QLD 4059, Australia; jyotsna.batra@qut.edu.au; 13Pathology, School of Medicine and Dentistry, Gold Coast Campus, Griffith University, Gold Coast, QLD 4222, Australia; a.lam@griffith.edu.au; 14Cancer Pharmacology Lab, AIRC Start Up Unit, Fondazione Pisana per La Scienza, 56017 Pisa, Italy

**Keywords:** machine learning, colorectal cancer, bioinformatics, biomarker, prognosis

## Abstract

**Simple Summary:**

Colorectal cancer (CRC) is among the leading causes of cancer-related deaths. Despite extensive efforts, a limited number of biomarkers and therapeutic targets have been identified. Therefore, novel prognostic and therapeutic targets are needed in the management of patients and to increase the efficacy of current therapy. The majority CRC patients follow the conventional chromosomal instability (CIN), which is started by several mutations such as APC, followed by genetic alterations in KRAS, PIK3CA and SMAD4, as well as the hyperactivation of pathways such as Wnt/TGFβ/PI3K. Although the underlying genetic changes have been well identified, the mutational signature of tumor cells alone does not enable us to subclassify tumor types or to accurately predict patient survival and suppression of those pathways have often not been effective in treatment. Our data showed some new genetic variants in *ASPHD1* and *ZBTB12* genes, which were associated with a poor prognosis of patients.

**Abstract:**

**Introduction:** Colorectal cancer (CRC) is a common cancer associated with poor outcomes, underscoring a need for the identification of novel prognostic and therapeutic targets to improve outcomes. This study aimed to identify genetic variants and differentially expressed genes (DEGs) using genome-wide DNA and RNA sequencing followed by validation in a large cohort of patients with CRC. **Methods:** Whole genome and gene expression profiling were used to identify DEGs and genetic alterations in 146 patients with CRC. Gene Ontology, Reactom, GSEA, and Human Disease Ontology were employed to study the biological process and pathways involved in CRC. Survival analysis on dysregulated genes in patients with CRC was conducted using Cox regression and Kaplan–Meier analysis. The STRING database was used to construct a protein–protein interaction (PPI) network. Moreover, candidate genes were subjected to ML-based analysis and the Receiver operating characteristic (ROC) curve. Subsequently, the expression of the identified genes was evaluated by Real-time PCR (RT-PCR) in another cohort of 64 patients with CRC. Gene variants affecting the regulation of candidate gene expressions were further validated followed by Whole Exome Sequencing (WES) in 15 patients with CRC. **Results:** A total of 3576 DEGs in the early stages of CRC and 2985 DEGs in the advanced stages of CRC were identified. *ASPHD1* and *ZBTB12* genes were identified as potential prognostic markers. Moreover, the combination of *ASPHD* and *ZBTB12* genes was sensitive, and the two were considered specific markers, with an area under the curve (AUC) of 0.934, 1.00, and 0.986, respectively. The expression levels of these two genes were higher in patients with CRC. Moreover, our data identified two novel genetic variants—the rs925939730 variant in *ASPHD1* and the rs1428982750 variant in *ZBTB1*—as being potentially involved in the regulation of gene expression. **Conclusions:** Our findings provide a proof of concept for the prognostic values of two novel genes—*ASPHD1* and *ZBTB12*—and their associated variants (rs925939730 and rs1428982750) in CRC, supporting further functional analyses to evaluate the value of emerging biomarkers in colorectal cancer.

## 1. Introduction

Colorectal cancer (CRC) is the second most common cause of cancer-related mortality [1], and its incidence is increasing despite the advances in the detection of prognostic and/or therapeutic targets. This is partly due to the limited number of therapeutic agents that have been identified. A high proportion of patients with CRC develop metastatic cancer(s) or become resistant to therapy. Therefore, novel prognostic biomarkers and new therapeutic targets that can help to assess the risk of developing CRC recurrence or increase the efficacy of current therapy are urgently needed.

Integrated analyses of multi-omics data provide useful insight into the pathogenesis of CRC and help to identify novel diagnostic and prognostic biomarkers. With the success of artificial intelligence technologies, machine learning (ML) is being used in healthcare. ML methods provide novel techniques of integration and analyzing omics for the discovery of novel biomarkers [2,3]. Hammad and collaborators [4] identified 105 differential expression genes (DEGs) using datasets from the Gene Expression Omnibus (GEO). Functional enrichment analysis revealed that these genes were enriched in cancer-related biological processes. The protein–protein interaction (PPI) network selected 10 genes, including *IGF1*, *MYH11*, *CLU*, *FOS*, *MYL9*, *CXCL12*, *LMOD1*, *CNN1*, *C3*, and *HIST1H2BO*, as hub genes. Support Vector Machine (SVM), Receiving Operating Characteristic (ROC), and survival analyses demonstrated that these hub genes can be considered potential prognostic biomarkers for CRC.

Maurya et al. [5] used Least Absolute Shrinkage and Selection Operator (LASSO) and Relief for feature selection from the Cancer Genome Atlas (TCGA) dataset and applied RF, K-Nearest Neighbor (KNN), and Artificial Neural Network (ANN) to check the accuracy of the models. The joint set of selected features between LASSO and DEGs was 38 genes, among which *VSTM2A*, *NR5A2*, *TMEM236*, *GDLN*, and *ETFDH* were correlated with the overall survival (OS) of patients with CRC and could be used as prognostic biomarkers. For example, Liu et al. [6] identified 16 lncRNAs as an immune-related lncRNA signature (IRLS) for predicting patients’ prognosis of CRC using machine learning-based integrated analysis. They performed further investigations to validate the application of IRLS in practice. The efficacy of immune-related lncRNA signature was validated using qRT-PCR on CRC tissues collected from 232 patients. A prospective cohort study, RECOMMEND (NCT05587452), aimed to assess the accuracy of a novel AI-based integrated analysis screening method for CRC and advanced colorectal adenomas using plasma multi-omics data.

Genome-wide association studies (GWAS) have already allowed significant progress in the understanding of the complex genetics behind the pathogenesis of CRC. There are at least three major molecular pathways that can lead to CRC, including the chromosomal instability pathway (characterized by aneuploidy or structural chromosomal abnormalities), chromosomal instability, and mutations (e.g., *APC*, *KRAS*, *PIK3CA*, *SMAD4*, or *TP53*). There is a growing body of evidence on targeting deregulated intracellular pathways, such as the hyperactivation of WNT–β-catenin, PI3K/Akt, or RAS signaling, although it has been shown that inhibiting these pathways has often not been effective in the clinical management of CRC [7,8,9,10]. Many patients with CRC had conventional chromosomal instability (CIN), which is started by several mutations such as APC, followed by genetic alterations in *KRAS*, *PIK3CA*, and *SMAD4*, as well as the hyperactivation of pathways such as Wnt/TGFβ/PI3K. Although the underlying genetic changes have been sufficiently identified, the mutational signature of tumor cells alone does not enable us to subclassify tumor types or to accurately predict patients’ survival, and the suppression of those pathways has often not been effective in treatment [11]. In this study, we attempted to develop and validate novel prognostic biomarkers based on ML-based integrated analysis as well as validation of novel candidate genes in two additional cohorts of CRC in DNA and RNA levels using whole exome sequencing (WES) and reverse transcription polymerase chain reaction (RT-PCR), respectively.

## 2. Materials and Methods

### 2.1. Data Sources and Data Processing

RNA-sequencing (RNA-seq) expression data and clinicopathological information were retrieved from The Cancer Genome Atlas (TCGA) database, which included 287 CRC tissue samples and 41 non-cancers tissue samples. In this study, RNA-seq data were obtained from TCGA-colorectal adenocarcinoma. Patients with colorectal cancer were classified into early-stage and late-stage. Early-stage CRCs were classified into three subgroups based on microsatellite instability (MSI) status: low MSI (MSI-L), high MSI (MSI-H), and MSI-stable (MSI-S). Late-stage CRCs were classified into two subgroups based on the therapeutic regimens (chemotherapy versus targeted therapy).

### 2.2. Patient’s Samples

Sixty-four CRCs were included in this study based on histological confirmation by two pathologists. All the eligible patients were chemotherapeutic naive patients treated at the Omid Hospital of Mashhad University of Medical Sciences. The study was approved by the local Hospital Ethic Committee of Mashhad University of Medical Sciences.

### 2.3. DNA-Seq and Whole Exome Sequencing

Data from the TCGA database were downloaded and prepared for further analysis in the R programming language. The data were downloaded in the Mutation Annotation Format (MAF). MAF is a standardized format used by TCGA for storing and analyzing various types of somatic mutations in cancer. The patients were divided into two groups: patients in the early stages (I, II) of CRC and patients in the advanced metastatic stage (IV). The first group consisted of 118 patients, while the second group consisted of 28 patients. MAF data belonging to each group is analyzed with the maftools package in R programming.

The genes with a significant *p*-value of less than 0.05 obtained from the survival analysis were combined with the whole exome sequencing data of TCGA for colorectal cancer. Then, the variants of the candidate genes obtained from sequencing data were analyzed using the Maftools package. Then, two candidate genes, ASPHD1 and ZBTB12, were further evaluated for their impact on gene expression using RegulomeDB and 3DSNP. Subsequently, the candidate genes were further confirmed in an additional cohort performed for the Whole Exome Sequencing (WES) data of 15 patients with CRC, as described previously.

### 2.4. Differential Gene Expression Analysis

Normalization was performed, while the PCA plots, volcano plots, heatmap, and karyoplote were represented by the R packages “ggplot2”, “heatmap”, and karyoploteR to visualize data. Significance analysis of differentially expressed genes (DEGs) was performed using DESeq2 in R software with the cutoff criteria of |log fold change | ≥ 1.5 and an adjusted *p*-value of <0.05.

### 2.5. Gene Set, Ontology, and Pathway Enrichment Analysis

The significant enrichment analysis of DEGs was assessed based on Gene Ontology (GO), Reactom, GSEA, and Human Disease Ontology (DO). GO analysis (http://www.geneontology.org/) is used for annotating genes and gene products and investigating the biological aspects of high-throughput genome or transcriptome data, including biological processes, cellular components, and molecular function. The Reactom database was used for the analysis of gene functions in biological signaling pathways. We set a *p*-value < 0.05 and a false discovery rate (FDR) < 0.05 as the statistically significant criteria to output. The whole transcriptome was employed for GSEA, and only gene sets with *p*-value < 0.05 and FDR q < 0.05 were set as statistically significant criteria. Statistical significance was set at an adjusted *p*-value of <0.05. Several R packages were utilized to perform enrichment analyses, including ReactomePA, enrichplot, clusterProfiler, and topGO.

### 2.6. Survival Analysis

The univariate/Cox proportional hazards regression model was used to identify DEGs that were significantly correlated with overall survival and assess the independent prognostic factors. R version 4.2.1 software was used to analyze the data.

### 2.7. Machine Learning Method

Two machine learning techniques were used, including the decision tree learning and deep learning. Deep learning models were applied to identify the effective factors. The significant variables obtained from the feature selection method (Weight by Correlation) were the final parameters in creating the model. The coefficient of correlation between variables is presented as a correlation matrix. The correlation coefficient is measured from –1 to 1; positive values represent that the variables are in the same direction, and negative correlations show the variables in opposite directions. The lack of correlation was displayed as 0.

### 2.8. Computational Workflow

Python3.7 was utilized for modeling. Parameters of epochs = 10, activation function = Rectifier, and learning rate = 0.01 were set in deep learning. The standard workflow was utilized as follows: Splitting the source data set into a training set and test set was performed to provide some independent evaluation levels. Subsequently, the model was optimized using the training data and then independently evaluated using the test data. In this study, a 70/30 train/test ratio was determined for the ML models. For each workflow, a model with the fixed optimal hyperparameter values was retrained on data and randomly sampled from the complete dataset. Machine learning method assessment was performed by 5 indicators, including accuracy, R2, MSE, and AUC.
Accuracy = (TP + TN)/(TP + TN + FP + FN)
where TP is true positive, FP is false positive, TN is true negative, and FN is false negative.
MSE (Mean Squared Error) = (1/n) × Σ (actual − forecast)2
where Σ represents a symbol that means “sum”, n is the sample size, actual is the actual data value, and the forecast is the predicted data value.
R2 (R-Squared) = 1 − Unexplained Variation/Total Variation

R2 is the coefficient of determination, and it tells you the percentage variation in y explained by x-variables. AUC (Area Under the Curve) represents the degree of separability and illustrates the capability of the model in distinguishing the classes.

### 2.9. Protein–Protein Interaction (PPI) Network

The STRING database (https://string-db.org/) was checked to find the relationship between the studied proteins obtained from DEG and the proteins that are directly or indirectly involved in the development of cancers. A minimum effective binding score of ≥0.4 was established. Genes with significant interactions were screened.

### 2.10. Kaplan–Meier Survival Curve

Kaplan–Meier survival curve comparison was conducted to measure the prognostic value of candidate genes in CRC using the log-rank test.

### 2.11. Receiver Operating Characteristic (ROC) Curve Analysis

Receiver operating characteristic (ROC) curves are a fundamental analytical tool for assessing diagnostic tests and identifying diagnostic biomarkers. ROC curve analysis evaluates the accuracy of a test to differentiate between diseased and healthy cases, thereby measuring the overall diagnostic performance [12]. A ROC curve and the area under the curve (AUC) were employed to determine the specificity, sensitivity, likelihood ratios, positive predictive values, and negative predictive values using the R package (pROC, version 1.16.2).

### 2.12. Quantitative Real-Time-PCR Validation

Total RNAs were extracted from tissues using a total RNA extraction kit according to the manufacturer’s protocol (Parstous, Tehran, Iran). RNA quantity and quality were assessed using a Nanodrop 2000 spectrophotometer (BioTek, USA EPOCH), and forty RNAs that passed the quality control were used for the next step. The RNAs were then reverse-transcribed into complementary DNA (cDNA) using a cDNA Synthesis Kit (Parstous, Tehran, Iran) according to the manufacturer’s instructions. Primers were designed (Forward Reverse: *ASPHD1*: AGTGGCTCACAATGGCTCC and AAGACAAAGTCGAGGGCCTG and *ZBTB12*: TTGCTCCTCTCCTGCTACACG and AACTGGCTGAGGGCATTCCG), and RT-PCR was performed via the ABI-PRISM StepOne instrument (Applied Biosystems, Foster City, CA, USA) using the SYBR green master mix (Parstous Co. Tehran, Iran). Gene expression data were standardized to glyceraldehyde 3-phosphate dehydrogenase (GAPDH) using a standard curve of cDNAs obtained from quantitative polymerase chain reaction (qPCR) Human Reference RNA (Stratagene, La Jolla, CA, USA).

## 3. Results

### 3.1. Whole Exome Sequencing

The Mutation Annotation Format (MAF) data were divided into two groups: patients in the early stages and advanced metastatic stage, as shown in Figure 1 and Figure 2, containing 118 and 28 patients, respectively. The MAF data were analyzed using the maftools package in R programming. Figure 1 and Figure 2 show different plots, including plot maf Summary, oncoplots, Transition and Transversions reports, Plotting VAF (Variant Allele Frequencies), Somatic Interactions reports, Drug–Gene Interactions, and Oncogenic Signaling Pathways to visualize the MAF distribution in a different group. As shown in Figure 1A and Figure 2A, in the early and late stages, missense mutations were more frequent than other mutations, and they were typically referred to as single-nucleotide polymorphism (SNP) types. Additionally, in both groups, 70–71% of patients had mutations in their APC or TP53 genes. Most of the variants are involved in Wnt/B-catenin _signaling, Genome integrity, and MAPK signaling (Figure 1B and Figure 2B). The clonal status of the most mutated genes can be estimated using the Variant Allele Frequencies plot; clonal genes usually have an average allele frequency of about 50% in pure samples. In the early stages of tumor development, TP53 was observed to have clonal status in the tumor tissue, while SMAD4, RYR4, and TP53 exhibit such a status in the late stages, as shown in Figure 1D and Figure 2D.

Somatic Interactions analysis indicated exclusive or co-occurrence (Figure 1E and Figure 2E). Mutually exclusive events happen in cancer when mutations in one gene prevent the occurrence of mutations in another gene. Co-occurring events, on the other hand, arise when mutations in two or more genes occur together more frequently than would be expected by chance. Determining mutually exclusive genes implies that these genes may participate in the same pathway or process, and there might be functional overlap between them. On the other hand, identifying genes that co-occur may indicate that they collaborate to facilitate the growth of tumors, or that their cumulative impact is essential for the development of cancer. The interaction between genes and drugs that target tyrosine kinase, transcription factor complex, DNA repair, and other related processes is illustrated in Figure 1F and Figure 2F. The involvement of mutated genes in colorectal cancer across different oncogenic signaling pathways, including RTK-RAS, Wnt, Hippo, Notch, and others, is demonstrated in Figure 1G and Figure 2G.

### 3.2. Gene Expression Profiling, Identification of DEGs, and Pathway Enrichment Analysis

We performed gene expression profiling in 287 CRC cases, analyzed by the DESeq2 package, according to the adjusted *p*-value of <0.05 and a |logFC| ≥ 1.5 (Appendix A). The PCA plots, volcano plots, and heat maps of each subgroup are shown in Figure 3 and Appendix A. Moreover, the gene expression of each subgroup, obtained from the DEG analysis was exhibited in the ideogram of chromosomes using the karyoploteR package (Figure 3C). Enrichment analysis showed that DEGs were significantly enriched in biological processes related to cancer progression. Based on GO analysis, the main biological processes involving the DEGs included ion homeostasis, inorganic cation transmembrane transport, and the regulation of hormone levels. In terms of cellular components, the DEGs were mostly enriched in the external encapsulating structure and extracellular matrix (ECM). In terms of molecular functions, the DEGs were linked by cation transmembrane transport activity, receptor regulator activity, signaling receptor activator activity, etc. (Figure 4A and Appendix A).

GSEA analysis showed that there was a relationship between identified DEGs and cell cycle, cell cycle checkpoint, DNA repair, mitotic nuclear division, cellular response to DNA damage stimulus, programmed cell death, epithelial cell differentiation, DNA-binding transcription factor activity, regulation of transcription by RNA polymerase II, Wnt signaling pathway, keratin filaments. According to the Reactom database analysis, DEGs were involved in GPCR signaling and its downstream signaling pathways, the regulation of Insulin-like growth factor (IGF), SLC-mediated transmembrane transport, the degradation of the extracellular matrix (ECM), collagen degradation, biological oxidation, and the activation of matrix metalloproteinases. (Figure 4B,C).

To further explore the prognostic value of emerging DEGs, we performed univariate Cox proportional hazards regression (Appendix A).

### 3.3. Machine Learning Analysis

The results of the ML analysis are shown in Table 1. The deep learning method achieved an accuracy of 97.14%, 97%, 98%, and 92% for predicting CRC in the MSI-H, MSS, chemotherapy, and targeted therapy subgroups, respectively, with AUC values of 1.0, 1.0, 1.0, and 0.88. This model had the best performance in the MSI-H and MSS subgroups. Then, 14 candidate genes were identified as novel genes which were dysregulated in both DNA and RNA levels. Also, the candidate genes and common genes resulting from the survival analysis were then displayed on a Venn diagram (Figure 4D and Appendix A). Following the visualization described in the MAF data analysis stage, 232 variants from 14 candidate genes related to survival were analyzed (Figure 5). Then, we confirmed the candidate genes in an additional cohort of our patients, which was detected by whole genome sequencing (WES) in 15 cases. Then, 11 genes emerged between the different cohorts, including ASPHD1, C2orf61, C6orf223, CADPS, CCDC150, DCAF4L1, MIA, NEK5, ONECUT3, PNPLA3, and TMEM145 (Appendix A).

### 3.4. The Prognostic Value of ZBTB12 and ASPHD1

Of note, RNA-seq data certified the dysregulation of candidate genes identified from Ml and DNA-seq and shortlisted *ZBTB12* and *ASPHD1* as the disease-associated genes (Figure 6). According to the Human Protein Reference Database, *ZBTB12* and *ASPHD1* interact with *HRAS*, *Ras-associated protein 1*, and *HRAS*, *PRRC2A*, *MSL3*, and *PIK3CA* (Figure 6A,B). The results of WES found nine genetic variants in *ASPHD1* and *ZBTB1* (Figure 6C,D). According to the RegulomeDB database and 3DSNP, the rs925939730 variant of the *ASPHD1* and rs1428982750 variant of the *ZBTB1* regulate gene expression and affect chromatin state in the colon and rectum (Appendix A). Moreover, the rs1428982750 variant was linked to *VARS* and *EHMT2* genes, and the rs925939730 variant was associated with the *MAZ* gene (Appendix A). The rs1428982750 variant of the *ZBTB12* gene had a score of 0.60906 for its role in gene expression regulation. Also, this variant affected the state of the chromatin transcription activity in the colon and rectum. Chromatin immunoprecipitation coupled with sequencing (CHIP-seq) results showed that the *ZBTB12* gene variant affects the binding site of transcription factors and various regulatory factors. (Appendix A). The rs925939730 variant of the *ASPHD1* gene had a score of 0.77967 for its role in regulating gene expression. Also, this variant affected the state of the chromatin transcription activity in the colon and rectum. CHIP-seq results showed that the *ASPHD1* gene variant affects the binding site of transcription factors and various regulatory factors. (Figure 6E). The results of the rs1428982750 variant of the *ZBTB12* gene in the 3DSNP database showed that the association of this variant with the regulatory factors of gene expression has a score of 58.4 (Appendix A). The different positions of this variant. The results of the rs925939730 variant of the *ASPHD1* gene in the 3DSNP database showed that the association of this variant with the regulatory factors of gene expression has a score of 59.7 (Appendix A).

ROC curve data was obtained by plotting the rate of sensitivity versus specificity. Also, Kaplan–Meier revealed that the overall survival of patients with cancer having low *ASPHD1* expression had higher overall survival (OS) than patients with cancer with high *ASPHD1* expression (*p* < 0.05). Similarly, cancers with high *ZBTB12* expression were associated with poor patient survival compared to cancers with low ZBTB12 expression (*p* < 0.05) (Figure 6F,G). As shown in Figure 6H and Table 2 and Table 3, *ASPHD1*, *ZBTB12*, and their combination were able to discriminate CRC with an area under the curve (AUC) of 0.948, 0.96, and 0.986, respectively. At the cutoff values of 0.863, 0.891, and 0.886, the sensitivities of *ASPHD1*, *ZBTB12*, and their combination were 0.878%, 0.861%, and 0.934%, respectively, with specificities of 1. The combination of *ASPHD1* and *ZBTB12* showed higher AUC and sensitivity than each of these candidate genes alone.

To further verify their values, the expression of these two candidate genes was evaluated in an additional cohort of CRC via qRT-PCR. The data showed a significantly higher expression of ASPHD1 and ZBTB12 in CRC tissues (*p* < 0.05) (Figure 6I).

## 4. Discussion

Colorectal cancer ranks as the third most common cause of cancer-related mortality [13]. Early diagnosis of this disease leads to more effective treatment, reduced treatment costs, reduced disease progression, and decreased morbidity and mortality. Since cancer is intimately linked to genetic alterations, pinpointing these changes is especially critical for early diagnosis. Implementing the right analyses of gene expression information can promote optimal treatment selection in the early stages of the development of various cancers. Identifying prognostic biomarkers and achieving diagnosis constitute a worthwhile tactic for disease management and care [14,15]. Artificial intelligence (AI) and deep learning (DL) are being widely adopted in medicine to enhance diagnosis, treatment, and research on diagnosing colorectal cancer (CRC) has followed this trend. DL is now integrated across CRC diagnostic approaches such as histopathology, endoscopy, radiology, and biochemical blood tests. By automating complex data analysis, DL allows for more precise CRC detection and characterization. Although AI adoption faces regulatory hurdles, it has the potential to optimize the diagnosis of CRC recurrence and personalized care by synthesizing diverse medical data and uncovering new insights. Overall, AI and DL are transforming the management of patients with CRC through improved diagnostic accuracy [16].

Our previous studies identified prognostic and diagnostic biomarkers in colorectal cancer and gastric cancer using RNA-seq analysis and machine learning [17,18,19]. In contrast to our previous study, the current study was designed based on an integrated two omics and deep learning approach to identify prognostic and diagnostic biomarkers in colorectal cancer (CRC) patients at different disease stages (early and metastatic). By combining multi-omics data and advanced computational methods, the present study provides novel insights into stratifying CRC patients based on genetic and expression profiles correlated with disease progression and outcomes. To the best of our knowledge, this is the first study showing the potential association of two genetic variants, rs1428982750 in *ZBTB12* and rs925939730 in *ASPHD1* genes, and the prognostic value of these genes in colorectal cancer. Bian Wu et al. used WES and RNA-seq to indicate prognosis prediction in patients with stage IV colorectal cancer. The results showed the following mutations in the genes: *APC*, *TP53*, *KRAS*, *TTN*, *SYNE1*, *SMAD4*, *PIK3CA*, *RYR2*. *BRAF* did not reveal any significant associations between the mutational status of those genes and patient prognosis [20]. Our study revealed that mutations in the genes *ZBTB12* and *ASPHD1* may serve as potential prognostic markers in patients. Specifically, we demonstrated that the mutational status of *ZBTB12* and *ASPHD1* was associated with clinical outcomes in the patient cohort examined. Chen et al. analyzed gene expression data from the GEO and TCGA databases and identified 10 hub genes with high diagnostic values based on ROC curve analysis. A nine-gene prognostic signature was also identified and shown to predict overall survival [21]. Importantly, we validated the expression of *ASPHD1* and *ZBTB12* genes through qPCR and their variants using whole exome sequencing in additional patient cohorts.

Data from the PPI network showed that *ASPHD1* is related to several proteins and genes such as *KIF22*, *INO80E*, *SEZ6L2*, and *DOC2A*, most of which are cancer-related. Kinesin family member 22 (KIF22) is a regulator of cell mitosis and cellular vesicle transport. It is involved in spindle formation and the movement of chromosomes during mitosis. The alteration of *KIF22* is associated with several cancers, including CRC. A previous study indicated that *KIF22* is upregulated in CRC samples and that KIF22 expression is correlated with tumors and the clinical stage of CRC. Moreover, the suppression of KIF22 inhibited cell proliferation and xenograft tumor growth [22].

*SEZ6L2* regulates cell fate by involving the transcription of type 1 transmembrane proteins. A study showed that *SEZ6L2* was significantly upregulated in CRC tissues, and this upregulation was associated with poor prognosis in patients with CRC [23]. Lastly, INO80E is involved in transcriptional regulation, DNA replication, and probably DNA repair. Therefore, we hypothesize that *ASPHD1* may play a critical role in the pathogenesis of CRC.

PRRT2 is also related to several kinds of human solid tumors [24]. The results of the Protein–protein interaction network demonstrated that *ZBTB12* is linked to numerous genes, including *HRAS*, *PIK3CA*, *MSL3*, and *PRRC2A*.

Phosphatidylinositol-4,5-bisphosphate 3-kinase (PI3K), an important kinase involved in the PI3K/AKT1/MTOR pathway, plays a crucial role in the growth and proliferation of various solid tumors, and *PIK3CA* is one of the most frequently mutated genes in CRC [25]. Harvey rat sarcoma viral oncogene homolog (HRAS) is involved in the activation of Ras protein signal transduction, and its mutations can be found in bladder and head and neck squamous cell carcinomas [26]. It has been shown that proline-rich coiled-coil2A (PRRC2A) takes part in tumorigenesis and immunoregulation. Recent studies have revealed that PRRC2A impacts pre-mRNA splicing and translation initiation [27]. In this context, several studies have demonstrated that there is a relationship between PRRC2A and several kinds of human cancers, such as hepatocellular carcinoma [28] and non-Hodgkin lymphoma [29].

Collectively, *ASPHD1* and *ZBTB12* are linked to multiple proteins and genes which are associated with cancer initiation and progression. Moreover, our results from WES analysis indicated that the rs925939730 variant of the *ASPHD1* gene and the rs1428982750 variant of the *ZBTB1* gene regulate gene expression and affect the chromatin state in the colon and rectum.

In addition, our findings demonstrated that there was an interaction between the rs1428982750 variant and *VARS* and *EHMT2* genes. Valyl-tRNA synthetase (VARS) was linked with CRC [30], breast cancer [31], and leukemia [30]. Euchromatic histone-lysine N-methyltransferase 2 (EHMT2) methylates histone H3 lysine 9 to generate heterochromatin and inhibit tumor suppressor genes [32]. Furthermore, the rs925939730 variant was associated with the *MAZ* gene. MAZ acts as a transcription factor that can be combined with c-MYC and GA box to regulate the initiation and termination of transcription. The deregulated expression of MYC-associated zinc finger protein (MAZ) is correlated with the progression of tumors such as colorectal adenocarcinoma [33], hepatocellular carcinoma [34], renal cell carcinoma [35], glioblastoma [36], breast carcinoma [37], and prostate adenocarcinoma [38]. Altogether, the *rs925939730* and *rs1428982750* gene variants of *ASPHD1* might be involved in gene expression and epigenetic regulation.

## 5. Conclusions

Our data show the prognostic value of *ASPHD1* and *ZBTB12* in CRC, warranting further investigations to validate their clinical potential as prognostic markers and predictive markers for colorectal cancer. Our study had some limitations and challenges, including the difficulty we experienced obtaining access to more patients for evaluating gene expression, carrying out functional studies, and analyzing other omics data to assess important pathways and biological processes in cancer. Expanding our omics approaches beyond just transcriptomics to also include proteomics, metabolomics, etc., would provide a more comprehensive understanding of the key mechanisms in cancer. Overcoming these limitations will be critical for future efforts to elucidate the complex molecular landscape of cancer and identify novel therapeutic targets or biomarkers.

## Figures and Tables

**Figure 1 cancers-15-04300-f001:**
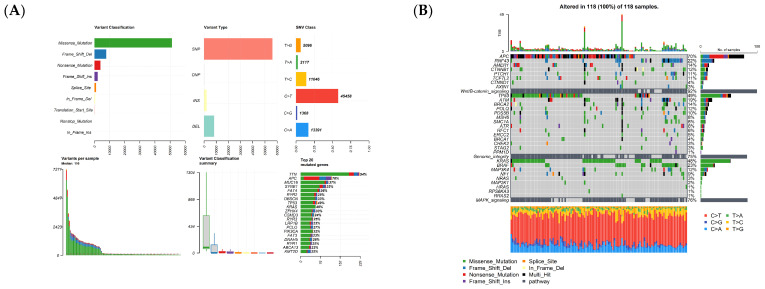
Visualization and summary of the analysis results of MAF data in the early-stage group (I, II stages) with the maftools package. (**A**) Bar and box plots display the frequency of different variants across samples (DEL: Deletion, INS: Insertion, SNP: Single-nucleotide polymorphism, ONP: Oligo-nucleotide polymorphism). (**B**) Oncoplots (note: variants annotated as Multi_Hit are genes that are mutated repeatedly within the same sample). (**C**) Transition and Transversion mutations (Ti: Transition; Tv: Transversions). (**D**) A boxplot of Variant Allele Frequencies. (**E**) Somatic Interactions show results of exclusive/co-occurrence event analysis. (**F**) Drug–gene interaction analysis based on the Drug–Gene Interaction database. (**G**) Oncogenic Signaling Pathways.

**Figure 2 cancers-15-04300-f002:**
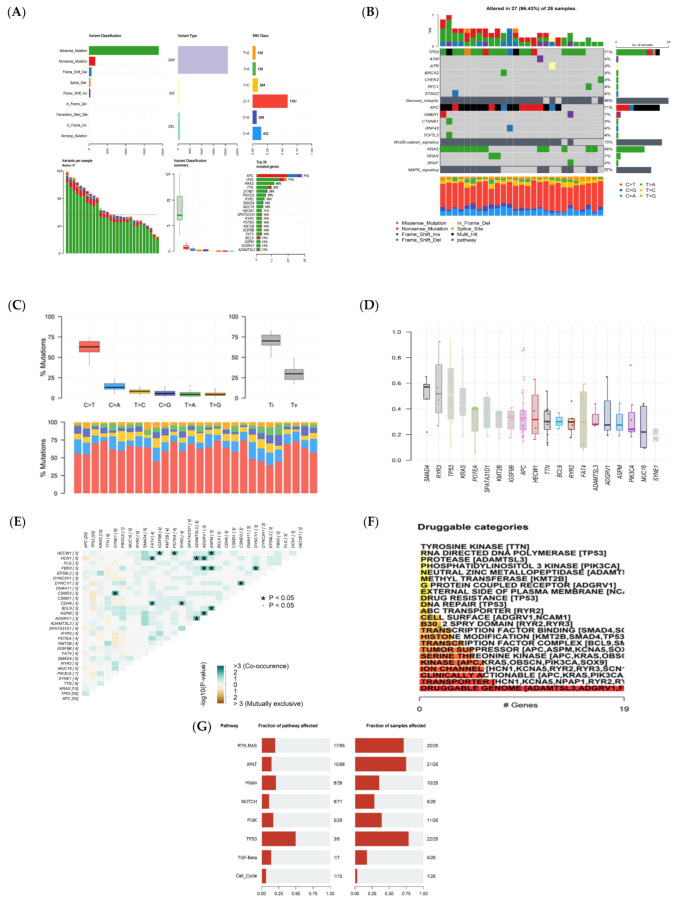
Visualization and summary of the analysis results of MAF data in the advanced-stage group (IV stage) with the maftools package. (**A**) Bar and box plots display the frequency of different variants across samples (DEL: Deletion, INS: Insertion, SNP: Single-nucleotide polymorphism, ONP: Oligo-nucleotide polymorphism). (**B**) Oncoplots (note: variants annotated as Multi_Hit are genes that are mutated repeatedly within the same sample). (**C**) Transition and Transversion mutations (Ti: Transition; Tv: Transversions). (**D**) Boxplot of Variant Allele Frequencies. (**E**) Somatic Interactions show the results of exclusive/co-occurrence event analysis. (**F**) Drug–gene interaction analysis based on the Drug–Gene Interaction database. (**G**) Oncogenic Signaling Pathways.

**Figure 3 cancers-15-04300-f003:**
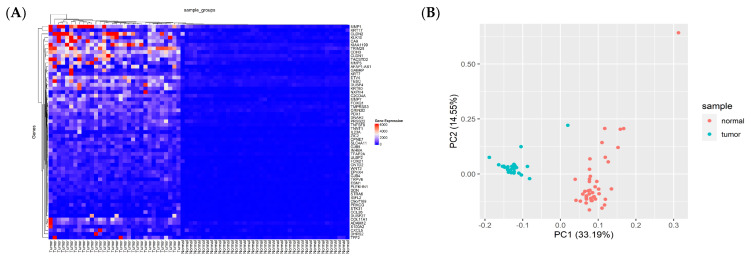
The results of the analysis of differentially expressed genes (DEGs) in colorectal adenocarcinoma (COAD) were generated using R software https://www.r-project.org/. (**A**) The heat map. (**B**) Principal component analysis (PCA). (**C**) karyoplot. (**D**) Volcano plot.

**Figure 4 cancers-15-04300-f004:**
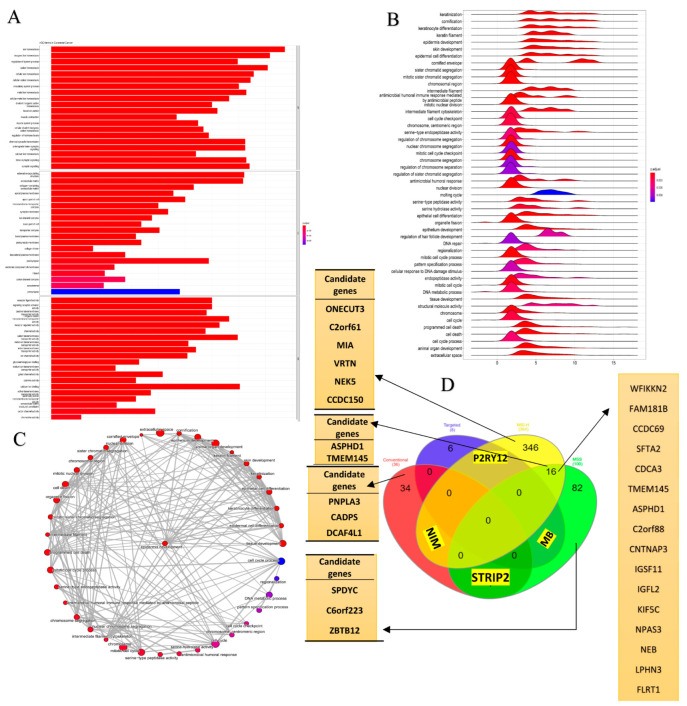
(**A**) Gene Ontology (GO), (**B**) GSEA functional annotation, and (**C**) Reactome functional pathways in colorectal adenocarcinoma (COAD). The *p*-value is less than 0.05 and is shown by the color. (**D**) A Venn diagram indicating the number of survival-related genes and the overlap between the different subgroups.

**Figure 5 cancers-15-04300-f005:**
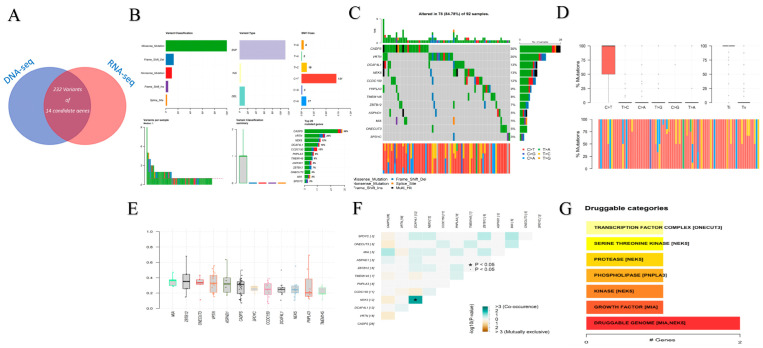
(**A**) A Venn diagram shows the count of variants for 14 candidate genes which are common between DNA-seq and RNA-seq analysis. (**B**) Bar and box plots displaying the frequency of different variants across samples. (**C**) Oncoplots. (**D**) Transition and Transversion mutations. (**E**) Boxplot of Variant Allele Frequencies. (**F**) Somatic Interactions show the results of exclusive/co-occurrence event analysis. (**G**) Drug–gene interaction analysis based on the Drug–Gene Interaction database.

**Figure 6 cancers-15-04300-f006:**
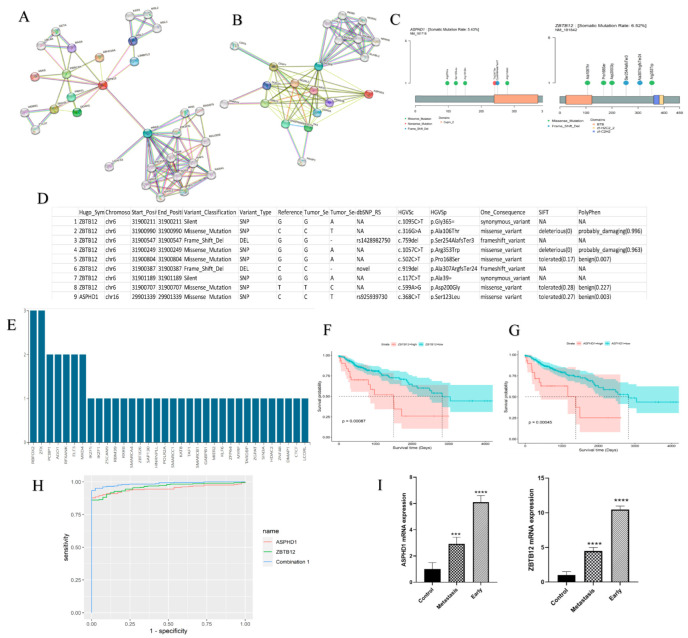
(**A**,**B**) Protein–protein interaction (PPI) network of the two genes (*ZBTB12*, *ASPHD1*) identified by survival analysis from STRING. (**C**,**D**) The different types of *ZBTB12* and *ASPHD1* variants, along with their respective alterations in the amino acid sequence on chromosomes, as well as the rate of somatic mutation. (**E**) CHIP-seq results have shown that variants of two genes (*ZBTB12*, *ASPHD1*) affect the binding site of transcription factors and various regulatory factors from the Regulome DB Database. (**F**,**G**) Kaplan–Meier plot of *ZBTB12* and *ASPHD1* with a prognostic value, *p*-value < 0.05. (**H**) ROC curve analysis revealed the biomarker potency of *ZBTB12* and *ASPHD1* individually and together using R 4.3.1’s combioROC package. (**I**) qRT-PCR results indicate that the expression levels of the two genes (*ZBTB12* and *ASPHD1)* are elevated in tumor tissue compared to non-neoplastic tissue. *** *p* > 0.01; **** *p* > 0.001.

**Table 1 cancers-15-04300-t001:** Results of machine learning analysis.

Subgroups	R2	AUC	MSE	RMSE	Accuracy	Prauc
MSI-H	0.99	1.0	1.95	0.0044	97.14%	1.0
MSI-S	0.99	1.0	0.0023	0.0489	97%	1.0
Receiving chemotherapy	0.95	1.0	0.0076	0.0876	98%	1.0
Receiving targeted therapies	0.64	0.88	0.0554	0.0235	92%	0.95

**Table 2 cancers-15-04300-t002:** The area under the curve (AUC) and a cut-off value of ASPHD1, ZBTB12, and their combination in CRC.

Biomarker	AUC	SE	SP	Cutoff	ACC	TN	TP	FN	FP	NPV	PPV
*ASPHD1*	0.948	0.878	1	0.863	0.893	41	252	35	0	0.539	1
*ZBTB12*	0.96	0.861	1	0.891	0.878	41	247	40	.	0.506	1
Combination	0.986	0.934	1	0.886	0.942	41	268	19	.	0.683	1

**Table 3 cancers-15-04300-t003:** Results for the ROC curve for ASPHD1, ZBTB12, and their combination in CRC.

Biomarker	Intercept	Coefficients	Degrees of Freedom	Null Deviance	Residual Deviance	AIC
*ASPHD1*	−10.37	Log (ASPHD1 + 1):3.032	327	247.2	136.3	140.3
*ZBTB12*	−22.345	Log (ASPHD1 + 1):5.165	327	247.2	118.3	122.3
Combination 1	−36.814	5,6,2	327	247.2	63.99	69.99

## Data Availability

The datasets are available upon request.

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
