# Peer review of "The Prognostic Value of ASPHD1 and ZBTB12 in Colorectal Cancer: A Machine Learning-Based Integrated Bioinformatics Approach"

_cancers, 2023, doi:10.3390/cancers15174300_

Round 1

Reviewer 1 Report

The abstract offers a thorough insight into a research study on the identification of genetic variants and differential expression genes (DEGs) in colorectal cancer (CRC). It outlines the methodology, comprising genome-wide DNA and RNA sequencing, survival analysis, and machine learning-based analyses, among others, and highlights key findings. While the study’s intention and results are evidently impactful, several areas require revision to enhance clarity and coherence:

The authors have provided an overview of the methods used in their study within the abstract. However, the Method section appears to be quite brief and lacks detailed explanations. Specifically, there seems to be an absence of comprehensive information concerning the hyperparameters of each method and technique applied. It is recommended that the authors expand the Method section to include a thorough description of the hyperparameters for each method used. This should encompass details such as the selection criteria for these hyperparameters, the rationale behind their values, and validation strategies implemented.

Author Response

Prof. Dr. Samuel C. Mok,

Thank you very much for your letter for the constructive comments made by yourself and the reviewers on our manuscript (cancers-2564797) entitled “The prognostic value of ASPHD1 and ZBTB12 in Colorectal Cancer: Machine Learning-Based Integrated Bioinformatics approach", submitted for consideration to be published as an original article in Cancers.

We are pleased to send you the manuscript carefully revised according to your advice as well as to the criticisms of the reviewers.

All authors are aware of and agree to the content of the manuscript.

We very much hope that the present version of the manuscript satisfactorily addresses all the observations made by you and the reviewers and meets the quality standard for publication in Cancers.

Thank you very much for your kind consideration.

Yours sincerely,

Amir Avan, [email protected]

REPLIES TO REVIEWERS’ COMMENTS

(Ref: Submission cancers-2564797)

Rev1

Open Review

Quality of English Language

( ) I am not qualified to assess the quality of English in this paper  
( ) English very difficult to understand/incomprehensible  
( ) Extensive editing of English language required  
( ) Moderate editing of English language required  
( ) Minor editing of English language required  
(x) English language fine. No issues detected  

Yes

Can be improved

Must be improved

Not applicable

Does the introduction provide sufficient background and include all relevant references?

(x)

( )

( )

( )

Are all the cited references relevant to the research?

(x)

( )

( )

( )

Is the research design appropriate?

( )

(x)

( )

( )

Are the methods adequately described?

( )

( )

(x)

( )

Are the results clearly presented?

(x)

( )

( )

( )

Are the conclusions supported by the results?

(x)

( )

( )

( )

Comments and Suggestions for Authors

The abstract offers a thorough insight into a research study on the identification of genetic variants and differential expression genes (DEGs) in colorectal cancer (CRC). It outlines the methodology, comprising genome-wide DNA and RNA sequencing, survival analysis, and machine learning-based analyses, among others, and highlights key findings. While the study’s intention and results are evidently impactful, several areas require revision to enhance clarity and coherence:

The authors have provided an overview of the methods used in their study within the abstract. However, the Method section appears to be quite brief and lacks detailed explanations. Specifically, there seems to be an absence of comprehensive information concerning the hyperparameters of each method and technique applied. It is recommended that the authors expand the Method section to include a thorough description of the hyperparameters for each method used. This should encompass details such as the selection criteria for these hyperparameters, the rationale behind their values, and validation strategies implemented.

Submission Date

01 August 2023

Date of this review

03 Aug 2023 03:42:00

Reply: We truly appreciate for your valuable comment on our manuscript to improve our paper. As we concur with the Reviewer’s request, these points are revised and added more explanation with blue highlights to the manuscripts.

Reviewer 2 Report

Here, the authors discuss the prognostic value of two genes, ASPHD1 and ZBTB12, in colorectal cancer. The authors used machine learning, bioinformatics, and experimental methods to identify and validate these genes as potential biomarkers. They also found two genetic variants that may regulate the expression of these genes.

Major points:

- The article is well-structured and follows the standard format of a scientific paper.

- The article uses multiple sources of data and methods to support the findings.

- The article provides novel insights into the molecular mechanisms and pathways involved in colorectal cancer.

- The article addresses a clinically relevant problem and proposes new prognostic markers for colorectal cancer.

Minor points:

- The article does not provide enough details on the machine learning models and their performance metrics. Please adress this in the methodology.

- The article does not discuss the limitations and challenges of the study, such as the sample size, the heterogeneity of the patients, and the potential confounding factors. Please adress this in the discussion.

- The article does not compare or contrast the findings with previous studies or existing literature.Please adress this in the discussion.

The article has some grammatical and typographical errors that need to be corrected. For example, “CRC” should be defined as “colorectal cancer” before using the abbreviation.  “The data from the TCGA database was downloaded and prepared for further analysis in 138 the R programming language.” should be “The data from the TCGA database were downloaded and prepared for further analysis in 138 the R programming language."

Author Response

Prof. Dr. Samuel C. Mok,

Thank you very much for your letter for the constructive comments made by yourself and the reviewers on our manuscript (cancers-2564797) entitled “The prognostic value of ASPHD1 and ZBTB12 in Colorectal Cancer: Machine Learning-Based Integrated Bioinformatics approach", submitted for consideration to be published as an original article in Cancers.

We are pleased to send you the manuscript carefully revised according to your advice as well as to the criticisms of the reviewers.

All authors are aware of and agree to the content of the manuscript.

We very much hope that the present version of the manuscript satisfactorily addresses all the observations made by you and the reviewers and meets the quality standard for publication in Cancers.

Thank you very much for your kind consideration.

Yours sincerely,

Amir Avan, [email protected]

REPLIES TO REVIEWERS’ COMMENTS

(Ref: Submission cancers-2564797)

REV2

Open Review

Quality of English Language

( ) I am not qualified to assess the quality of English in this paper  
( ) English very difficult to understand/incomprehensible  
( ) Extensive editing of English language required  
( ) Moderate editing of English language required  
(x) Minor editing of English language required  
( ) English language fine. No issues detected  

Yes

Can be improved

Must be improved

Not applicable

Does the introduction provide sufficient background and include all relevant references?

(x)

( )

( )

( )

Are all the cited references relevant to the research?

(x)

( )

( )

( )

Is the research design appropriate?

(x)

( )

( )

( )

Are the methods adequately described?

( )

(x)

( )

( )

Are the results clearly presented?

(x)

( )

( )

( )

Are the conclusions supported by the results?

(x)

( )

( )

( )

Comments and Suggestions for Authors

Here, the authors discuss the prognostic value of two genes, ASPHD1 and ZBTB12, in colorectal cancer. The authors used machine learning, bioinformatics, and experimental methods to identify and validate these genes as potential biomarkers. They also found two genetic variants that may regulate the expression of these genes.

Major points:

- The article is well-structured and follows the standard format of a scientific paper.

- The article uses multiple sources of data and methods to support the findings.

- The article provides novel insights into the molecular mechanisms and pathways involved in colorectal cancer.

- The article addresses a clinically relevant problem and proposes new prognostic markers for colorectal cancer.

Minor points:

- The article does not provide enough details on the machine learning models and their performance metrics. Please adress this in the methodology.

 Reply: We appreciate very much the constructive comments of this reviewer on our manuscript. We have revised this issue.

- The article does not discuss the limitations and challenges of the study, such as the sample size, the heterogeneity of the patients, and the potential confounding factors. Please adress this in the discussion.

 Reply: In agreement with these valuable points, We corrected them in the revised version of our paper.

- The article does not compare or contrast the findings with previous studies or existing literature.Please adress this in the discussion.

 Reply: In agreement with this valuable comment, we revised our text as requested.

Comments on the Quality of English Language

The article has some grammatical and typographical errors that need to be corrected. For example, “CRC” should be defined as “colorectal cancer” before using the abbreviation.  “The data from the TCGA database was downloaded and prepared for further analysis in 138 the R programming language.” should be “The data from the TCGA database were downloaded and prepared for further analysis in 138 the R programming language."

Reply: : Thank you so much for your  comment. Sorry for our mistakes; it was corrected.

Reviewer 3 Report

I was glad to review the work of the authors regarding this very interesting article on the Prognostic value of ASPHD1 and ZBTB12 in Colorectal Cancer. The manuscript is well-written and the incorporated tables and figures make the study easy to follow.

I strongly recommend acceptance for publication of the paper after minor changes.

1) I would like a brief discussion on the entire application spectrum of deep learning in all diagnostic tests regarding colon cancer, from endoscopy and histologic examination to medical imaging and screening serologic tests.

Consider citing the recently published article:

https://www.termedia.pl/The-role-of-deep-learning-in-diagnosing-colorectal-cancer,41,51103,0,1.html

General comments
The spelling and punctuation are very good. No issues were detected.
Abstract
The abstract is concise. All the necessary information about the study is included.

Background
- The information provided in the introduction is important for the comprehension of the article.
- The objective of the study is clearly mentioned.
Methods
- The methods are sufficiently explained by the authors.

Results
- The results are presented in a very extensive way.
- The table is really helpful and necessary for the completion of the authors' work.
Discussion
- The discussion is of great quality and includes updated data.
- The authors inform the reader about the study's limitations.
Conclusion
From the presented data, the conclusion is complete and represents the work that the authors did.

Author Response

Prof. Dr. Samuel C. Mok,

Thank you very much for your letter for the constructive comments made by yourself and the reviewers on our manuscript (cancers-2564797) entitled “The prognostic value of ASPHD1 and ZBTB12 in Colorectal Cancer: Machine Learning-Based Integrated Bioinformatics approach", submitted for consideration to be published as an original article in Cancers.

We are pleased to send you the manuscript carefully revised according to your advice as well as to the criticisms of the reviewers.

All authors are aware of and agree to the content of the manuscript.

We very much hope that the present version of the manuscript satisfactorily addresses all the observations made by you and the reviewers and meets the quality standard for publication in Cancers.

Thank you very much for your kind consideration.

Yours sincerely,

Amir Avan, [email protected]

REPLIES TO REVIEWERS’ COMMENTS

(Ref: Submission cancers-2564797)

REV3

Open Review

Quality of English Language

( ) I am not qualified to assess the quality of English in this paper  
( ) English very difficult to understand/incomprehensible  
( ) Extensive editing of English language required  
( ) Moderate editing of English language required  
( ) Minor editing of English language required  
(x) English language fine. No issues detected  

Yes

Can be improved

Must be improved

Not applicable

Does the introduction provide sufficient background and include all relevant references?

(x)

( )

( )

( )

Are all the cited references relevant to the research?

( )

(x)

( )

( )

Is the research design appropriate?

(x)

( )

( )

( )

Are the methods adequately described?

(x)

( )

( )

( )

Are the results clearly presented?

(x)

( )

( )

( )

Are the conclusions supported by the results?

(x)

( )

( )

( )

Comments and Suggestions for Authors

I was glad to review the work of the authors regarding this very interesting article on the Prognostic value of ASPHD1 and ZBTB12 in Colorectal Cancer. The manuscript is well-written and the incorporated tables and figures make the study easy to follow.

I strongly recommend acceptance for publication of the paper after minor changes.

1) I would like a brief discussion on the entire application spectrum of deep learning in all diagnostic tests regarding colon cancer, from endoscopy and histologic examination to medical imaging and screening serologic tests.

Consider citing the recently published article:

https://www.termedia.pl/The-role-of-deep-learning-in-diagnosing-colorectal-cancer,41,51103,0,1.html

Reply: we truly appreciate the comments of the rviewer on our manuscript, As we concur with the Reviewer’s request, these points are revised

General comments
The spelling and punctuation are very good. No issues were detected.
Abstract
The abstract is concise. All the necessary information about the study is included.

Background
- The information provided in the introduction is important for the comprehension of the article.
- The objective of the study is clearly mentioned.
Methods
- The methods are sufficiently explained by the authors.

Results
- The results are presented in a very extensive way.
- The table is really helpful and necessary for the completion of the authors' work.
Discussion
- The discussion is of great quality and includes updated data.
- The authors inform the reader about the study's limitations.
Conclusion
From the presented data, the conclusion is complete and represents the work that the authors did.